# Fast counting with tensor networks

Stefanos. Kourtis[1⋆], Claudio Chamon[1],
Eduardo R. Mucciolo[2] and Andrei E. Ruckenstein[1]

**1** Physics Department, Boston University, Boston, Massachusetts 02215, USA
**2** Department of Physics, University of Central Florida, Orlando, Florida 32816, USA

⋆ kourtis@bu.edu

## Abstract

We introduce tensor network contraction algorithms for counting satisfying assignments of constraint satisfaction problems (#CSPs). We represent each arbitrary #CSP formula as a tensor network, whose full contraction yields the number of satisfying assignments of that formula, and use graph theoretical methods to determine favorable orders of contraction. We employ our heuristics for the solution of #P-hard counting boolean satisfiability (#SAT) problems, namely monotone #1-in-3SAT and #Cubic-Vertex-Cover, and find that they outperform state-of-the-art solvers by a significant margin.

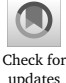

# 1  Introduction

Constraint satisfaction problems (CSPs) have both a prominent function in the fundamental understanding of computational complexity and a vast reach in applications across diverse fields of science. The classification of computations in terms of complexity implies the existence of efficient algorithms for problems classified as "tractable". In contrast, many CSPs are classified as "hard": it is expected that the amount of computational effort it takes for *any* algorithm to solve an instance of the problem scales superpolynomially with the size of the instance. This has led to decades of incremental refinement of algorithmic tools that can treat them at an acceptable computational cost [1]. A large number of these hard problems are of great practical importance. New algorithms for faster solution of a CSP can thus have far reaching consequences.

The search for such algorithms has led to confluences of computer science and computational many-body physics, with notable examples like the FKT algorithm for counting the perfect matchings of planar graphs [2–4] and survey propagation for the solution of boolean satisfiability problems [5–7]. In the study of computational problems, one is often interested in obtaining exact solutions. This is particularly pertinent to solution of problems that cannot be approximated to a desired accuracy with any substantial advantage in computational effort compared to solving the problem exactly. On the other hand, the majority of numerical many-body techniques trade the exact solution of a problem, which typically requires computation times that scale exponentially with the problem size, for an advantageous polynomial-time procedure that yields adequate approximations.

In this work, we follow the opposite route: we take a highly efficient numerical tool from many-body physics, namely tensor networks, and trade the benefit of polynomial scaling for the capability of obtaining the exact solution of hard computational problems, focusing in particular on counting satisfying assignments of boolean satisfiability (SAT) instances. In the original context of condensed matter physics, a tensor network is a representation of the classical or quantum partition function of a many-body system. In this language, the evaluation of the partition function amounts to taking the trace over all tensor indices. This representation formed the basis of powerful techniques for solving challenging problems of strongly correlated systems [8,9]. Tensor networks have since proliferated beyond interacting particle systems [10–12].

The problem of taking the tensor trace of an arbitrary tensor network belongs to the complexity class #P-hard [13]. So far, this has been circumvented in mainly two ways. The first is to impose restrictions on network structure, tensor entries, or both, and exploit these to devise subexponential-time exact contraction algorithms [14–21]. The second and so far most fruitful approach is to find an efficient scheme for accurate numerical approximation of the tensor trace. The idea is to perform local operations that *compress* the crucial information into subsets of the dimensions of a tensor, and truncate the remainder. Since in condensed matter physics one typically deals with systems defined on a lattice instead of an arbitrary graph, it is natural to establish a coarse-graining procedure inspired by the renormalization group apparatus [22–30]. The resulting methods approximate the tensor trace in time that scales polynomially with the number of degrees of freedom. Recently, coarse-graining algorithms that trade the polynomial scaling for *exact* computation of the tensor trace were introduced and successfully applied to the study of vertex models encoding computational problems [31].

CSPs are commonly defined on random graphs. Even though some CSPs — in particular, decision (SAT) and counting (#SAT) problems [32–34] — have been formulated as tensor networks, no practical strategies for their efficient contraction exist. This is partly because it is nontrivial to define coarse-graining protocols in arbitrary graphs, but also because there is frequently no obviously advantageous order of contraction. Even though problems defined on

arbitrary graphs can be embedded into lattices, this embedding often incurs large overheads in terms of ancillary degrees of freedom, which can in turn translate to undesirable computational penalties.

Here we introduce tensor network contraction methods for problems defined on *arbitrary* graphs. Our main goal is to devise strategies for finding favorable orders of contraction. Unlike most existing approaches, we forgo completely any compression scheme, approximate or exact. Using tools from graph theory, we develop contraction algorithms based on graph partitioning that are provably subexponential for certain classes of graphs with upper bounded maximum degree. We then extend the scope to broader classes of graphs and focus on random regular graphs as an example. We apply our algorithms to two #P-complete problems, namely, monotone #1-IN-3SAT and #CUBIC-VERTEX-COVER, two #P-complete #SAT problems defined on random 3-regular graphs, and show that our heuristics indeed yield favorable contraction sequences that enable us to outperform state-of-the-art #SAT solvers by a large margin.

We begin the presentation with a description of the tensor network contraction problem in Sec. 2.1. In Sec. 2.2 we cast the problem into a graph theoretic language. We use this vocabulary to prove that full contraction of tensors defined on graphs with sublinear separators can be performed in subexponential time, using graph partitioning as a tool. In Sec. 3.1 we detail numerical heuristics for full contraction of arbitrary networks of tensors. In Sec. 3.2 we test our algorithms on #CUBIC-VERTEX-COVER and show that it compares favorably to modern #SAT solvers. We conclude with a summary and outlook in Sec. 4

## 2 Definitions and preliminary observations

### 2.1 Tensor networks and boolean satisfiability

Consider a set of $N$ variables, denoted as $\{i\} = \{i_1, i_2, \ldots, i_N\}$, each of which is defined over a finite and discrete domain with at most $D$ elements. Variables are assumed to interact with one another. Interactions between a subset $\{s\} \subseteq \{i\}$ of variables can be expressed by a multivariate function $E_{\{s\}}(i)$, where the subscript indicates that the function depends only on the entries of the configuration vector (or "state") $i$ that correspond to the variables in $\{s\}$. $E_{\{s\}}(i)$ can be thought of as an energy cost penalizing "less compatible" variable configurations. One can then define the totality of interactions as

$$E(i) = \sum_{\{s\}} E_{\{s\}}(i). \tag{1}$$

In statistical mechanics, one defines the partition function as

$$Z = \sum_{i} \exp[-\beta E(i)] = \sum_{i} \prod_{\{s\}} T_{\{s\}}, \tag{2}$$

where $\beta$ is the inverse temperature. The objects $T_{\{s\}} = \exp(E_{\{s\}})$ are also multivariate functions of discrete variables — or *tensors* — and they fully encode the underlying interactions or constraints between variables.

With a definition that ensures $E(i) \geq 0 \ \forall \ i$, the Boltzmann factors $\exp[-\beta E(i)]$ express the occurrence probability of a configuration $i$ of "energy" $E(i)$ relative to the probability of a zero-energy state. When $\beta \to \infty$, $Z$ simply counts the number of zero-energy states and its calculation amounts to solving a combinatorial problem defined by the constraints imposed by $E_{\{s\}}$, or equivalently by $T_{\{s\}}$. Note that a naive exhaustive search for these states over all possible configurations requires an exponential number of operations.

This formulation allows for directly casting a variety of decision and counting CSPs as tensor networks. The main goal is to perform the summation on the right-hand side of Eq. (2),

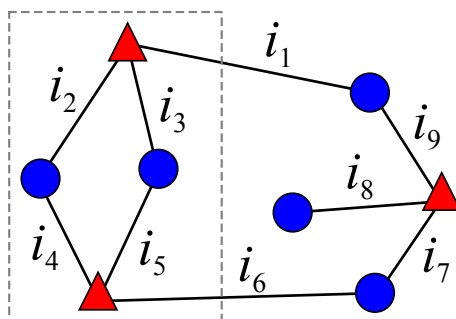

Figure 1: Representation of a CSP instance as a graph. Circles (triangles) represent variables (clauses). The dashed rectangle delineates the tensor contraction example discussed in the text.

which is referred to as *tensor trace* or *full contraction* and yields the solution of the under-lying instance. In general, tensor entries can take either continuous (e.g., $\mathbb{R}$, $\mathbb{C}$) or discrete (e.g., $\mathbb{Z}$, $\mathbb{N}$, $\mathbb{B}$) values. For simplicity and concreteness, below we focus on counting satisfying assignments in boolean satisfiability (SAT) problems. Tensor entries will hence be nonnega-tive integers. We stress, however, that all results are straightforwardly generalizable to CSPs beyond the boolean domain.

A SAT problem is the problem of deciding whether a logic formula built from a set of boolean variables $\{x\} = \{x_1, x_2, \ldots, x_n\}$ and the operators $\wedge$ (conjunction), $\vee$ (disjunction), and $\neg$ (negation) evaluates to TRUE, i.e., is *satisfiable*. In the so-called *conjunctive normal form*, variables and negations thereof — collectively called *literals* — are combined in *clauses*, i.e., disjunctions of literals, $m$ of which are in turn combined conjunctively to form the SAT formula. The general SAT problem defined in this manner, as well as many of its special cases, is NP-complete. In particular, when clauses are restricted to contain exactly $k$ variables each, the corresponding problem is called $k$SAT and is also NP-complete for all $k \geq 3$. The corresponding counting problems (#SAT) — determining how many satisfying assignments, if any, a SAT formula has — are at least as hard as their decision counterparts, and belong to the class #P-complete. In fact, $\#k$SAT is #P-complete for any $k \geq 2$.

Instances of $(\#)$SAT problems can be straightforwardly represented as tensor networks [32–34]. Each variable and clause is encoded into a tensor $T_{i_1 i_2 \ldots i_d}$, where $d$ is the rank of the ten-sor and all indices are boolean. Clause tensors reflect the underlying boolean operations on variables. For example, OR tensors are of the form

$$T^{\text{OR}}_{i_1 i_2 \ldots i_d} = \begin{cases} 0, & \text{if } i_1 = i_2 = \cdots = i_d = 0 \\ 1, & \text{otherwise} \end{cases}, \tag{3}$$

where each index labels a bond and represents a variable appearing in the clause. If a variable is to appear negated in a clause, the values of the corresponding boolean index of the clause tensor are reversed. Reversed indices are overlined, e.g., $T_{i_1 \bar{i}_2 i_3}$. Since variables can appear in more than one clauses, we need to be able to "replicate" the same index across clause tensors. This is achieved with COPY tensors of the form

$$T^{\text{COPY}}_{i_1 i_2 \ldots i_d} = \begin{cases} 1, & \text{if } i_1 = i_2 = \cdots = i_d \\ 0, & \text{otherwise} \end{cases}, \tag{4}$$

which indeed just "copies" the value of a variable across all indices.

With these definitions, tensor entries reveal how many assignments — if any — of the participating variables satisfies the underlying clause. For example, $T^{\text{OR}}_{000} = 0$ means that an

assignment in which all variables participating in a 3-variable OR clause are 0 is unsatisfiable, whereas $T_{001}^{\mathrm{COPY}} = 0$ means that errors in copying a variable are disallowed.

Common indices of two tensors can be "traced over", which results in the *contraction* of the two tensors into one. A tensor obtained from the contraction of two or more tensors still encodes the number of satisfiable assignments for each combination of values of the remaining indices. Consider, for example, the contraction

$$T_{i_1 i_6} = \sum_{i_2, i_3, i_4, i_5} T_{i_1 i_2 i_3}^{\mathrm{OR}} T_{i_2 i_4}^{\mathrm{COPY}} T_{i_3 i_5}^{\mathrm{COPY}} T_{i_4 i_5 i_6}^{\mathrm{OR}} = \begin{pmatrix} 3 & 3 \\ 3 & 4 \end{pmatrix}, \tag{5}$$

shown schematically in Fig. 1. This simply verifies that there are 4 satisfiable assignments of $i_2 = i_4$ and $i_3 = i_5$ if $i_1 = i_6 = 1$ and only 3 otherwise, leading to $T_{00} = T_{01} = T_{10} = 3$ and $T_{11} = 4$.

As discussed above, the full trace (2) of a tensor network yields the total number of configurations that satisfy all constraints. Therefore, if a tensor network encodes an instance of a SAT problem, its full contraction yields the solution of the corresponding #SAT counting problem.

A tensor network contraction algorithm is a method that evaluates the tensor trace numerically, i.e., contracts all bonds sequentially until a single, zero-rank tensor — a scalar — is obtained. If we allow only for contractions of tensors, then the maximum tensor rank in the network increases as the number of tensors decreases. A linear increase in the maximum rank means an exponential increase in both the cost of subsequent contractions and the memory required to store the tensor network. This means that the *order* of contractions can play a crucial role in the overall performance of a tensor network contraction algorithm.

The resource that defines the computation time is the tensor rank: a successful contraction strategy must keep the maximum tensor rank as low as possible throughout the contraction sequence. We note that the notion of *bond dimension*, commonly discussed in the physics literature, is fully equivalent to the tensor rank in the present context, since we do not ever perform any decomposition or truncation of tensor entries. The bond dimension is therefore fully determined by the number of "legs sticking out" of each tensor, i.e., the *degree* of the corresponding vertex in the network. As we detail in Sec. 2.2, efficient contraction is essentially a problem in algorithmic graph theory. Using methods of graph partitioning and community detection to tackle this problem, in Sec. 3 we will introduce practical tensor network contraction algorithms that construct favorable contraction sequences.

## 2.2 Graph theory and contraction complexity

The tensor network contraction problem is fundamentally a *graph modification* problem. Graph modification problems are an important aspect of algorithmic graph theory. Decision instances of such problems are typically stated as follows: given an input graph $G$, decide whether it is possible to obtain another graph $G'$ by performing a finite number of graph modification operations, such as vertex and edge additions or deletions, subject to some constraints. Problems of this kind typically tend to be NP-hard [35, 36]. The corresponding optimization problems, i.e., finding a sequence of graph modification operations that leads from $G$ to $G'$ ensuring a minimum number of violated constraints, are at least as hard as their decision counterparts.

To define the problem, let $G = (V, E)$ denote a graph with vertex set $V$ and edge set $E = \{uv : u, v \in V\}$, where we have denoted an edge connecting vertex $u$ to vertex $v$ as $uv$. The numbers of vertices and edges are written as $|V|$ and $|E|$, respectively. The *degree* $\deg v_i$ of a vertex $v_i \in V$ is the number of vertices it is adjacent to, so that $\sum_{i=1}^{|V|} \deg v_i = 2|E|$. We

Figure 2: Cartoon "snapshots" of a contraction sequence based on a separator hierarchy. Circles designate vertices of a graph and black lines are separators; edges are not shown. Circle size is proportional to vertex degree and thicker lines are separators higher in the hierarchy. Starting from the leftmost panel, edges of the shortest separators in the hierarchy are contracted, yielding the graph in the next panel.

also denote the *minimum and maximum degrees* of $G$ as $\delta(G)$ and $\Delta(G)$, respectively. A graph is *connected* if there is a path from any vertex to any other vertex. A graph is called *d-regular* if $\deg v_i = d \ \forall \ v_i \in V$. A graph is called *bipartite* if it can be divided into two components, such that each vertex belonging to one component is only connected to vertices belonging to the other. A graph is called *planar* if it can be embedded in the plane, i.e., it can be drawn on a compact two-dimensional manifold of genus zero in a way that its edges intersect only at their endpoints.

Below we will only consider the graph modification operation called *edge contraction*. Edge contraction is an operation that removes an edge $uv$ from a graph and replaces the vertices $u$ and $v$ by a new vertex $w$, such that all edges previously incident upon $u$ and $v$ (apart from $uv$, which has been removed) are now incident upon $w$. Contracting one or more edges of a graph $G$ generates *minors* of $G$. A minor of a graph $G$ is a graph that can be obtained from $G$ by any sequence of edge deletions, vertex deletions, and edge contractions. Below we implicitly assume that multiple edges between adjacent vertices are all contracted at once, so that minors obtained from edge contractions have no loops, i.e., there are no edges that connect a vertex to itself. *Full contraction* is thus the operation of reducing a graph to a single vertex via repeated edge contractions.

With the above definitions, the problem that motivates this work can be stated as follows:

BOUNDED DEGREE FULL CONTRACTION

*Input:* A graph $G$ and an integer $D$.

*Question:* Is there an edge contraction sequence that reduces $G$ to a single vertex, ensuring that every minor $H$ of $G$ generated in the contraction sequence has maximum degree $\Delta(H) < D$?

Without any restriction on the input graph $G$, BOUNDED DEGREE FULL CONTRACTION can be shown to be NP-complete for any fixed $D \geq 2$ [37]. The optimization problem of finding the contraction sequence that minimizes $D$ — which is equivalent to finding the optimal contraction sequence for the evaluation of the tensor trace (2) — is at least as hard as the decision problem BOUNDED DEGREE FULL CONTRACTION.

Our goal in this work is to show that we can obtain near-optimal contraction sequences despite the NP-hardness of BOUNDED DEGREE FULL CONTRACTION, and that this can have important practical consequences for the solution of constraint satisfaction problems using tensor networks. We first restrict the problem to planar graphs and derive a straightforward analytical result. This result then provides the impetus for the numerical heuristics applied to examples of #P-complete problems in Sec. 3.

It is well known that many NP-complete problems defined on graphs can be solved in subexponential time, when the graphs considered are planar. Here we will translate this into

the language of tensor networks and #P-complete problems. The general algorithm for the solution of a problem defined on a planar graph is to recursively partition the graph by finding *vertex separators* and employing dynamic programming methods. A vertex separator is a subset $S \subset V$, such that if $S$ and all the edges incident to it were deleted, then $G$ would split into two disconnected *induced subgraphs*. The planar separator theorem states that any planar graph on $|V|$ vertices has a separator $S$ that contains $O(|V|^{1/2})$ vertices and separates the graph into subgraphs $A$ and $B$, each of which has at most $2|V|/3$ vertices. Such planar separators can be found in linear time [38]. Correspondingly, there is a set of edges $C = \{uv \in E \mid u \in S, v \in A\}$, called an *edge separator* or *cut-set*. If a graph has maximum degree $\Delta$, then $|C|$ is $O((\Delta|V|)^{1/2})$ [39]. One can go one step further and construct *separator hierarchies*, by recursively separating $A$ and $B$ as described above. Such hierarchies can be calculated in linear time [40]. This procedure eventually distributes the vertices of the original lattice into $O(|V|)$ subgraphs, each of size $O(1)$.

Two immediate corollaries of the planar separator theorem and its generalizations follow:

**Corollary 1.** *Let $G$ be a planar graph with maximum degree $\Delta$. A sequence of edge contractions that fully contracts $G$, ensuring that the maximum degree of every minor $H$ of $G$ generated during the contraction sequence is at most $O((\Delta|V|)^{1/2})$, always exists and can be found in linear time.*

To prove this, find an edge separator hierarchy in linear time that partitions $G$ into $|V|$ subgraphs of size 1. Then calculate the tensor trace by recursively contracting all edges belonging to the *smallest* separators in the hierarchy in at most $O(|V|)$ steps in total, as illustrated in the cartoon of Fig. 2. At each step, the set of edges incident to a vertex belong to $O(1)$ separators, and hence they are at most $O((\Delta|V|)^{1/2})$. □

It is straightforward to generalize this corollary to other classes of graphs with sublinear separators, such as fixed-genus graphs [41], or more generally graphs with polynomial expansion [42].

In the context of tensor networks, Corollary 1 in turn has the following immediate consequence:

**Corollary 2.** *Let $\mathcal{T}_N$ be a planar tensor network of $N$ tensors with maximum rank upper bounded by a constant $\Delta$ and maximum dimension of a single index upper bounded by a constant $D$. Then $\mathcal{T}_N$ can be fully contracted in time $D^{O((\Delta N)^{1/2})}$.*

As in Corollary 1, find an edge separator hierarchy in time $O(\Delta N)$. Full contraction of the tensor network then requires at most $\Delta N$ contractions of two tensors, each of size at most $O(D^{(\Delta N)^{1/2}})$. Contraction of two tensors, each with a total number of elements $L$, costs time at most $O(L^2)$, and hence the total cost of contraction scales as $D^{O((\Delta N)^{1/2})}$. □

This observation implies that instances of any #CSP, defined on planar graphs with maximum degree $\Delta$ and with variables that can take at most $D$ values, can be solved with tensor networks in subexponential time *irrespective of what the entries of the tensors are*. On the other hand, even stronger results can be obtained for tensor networks of planar or fixed-genus #CSP instances if restrictions are imposed on tensor values [15–17,20]. In physics terms, Corollary 1 may be intuitively understood as a form of "area law" for classical systems [43]. In complexity theory, it is a manifestation of the so-called "square root phenomenon".

Corollary 2, like Corollary 1, also straightforwardly generalizes to classes of graphs with sublinear separators. In contrast, without any restriction on graph properties, the planar separator theorem and its generalizations do not hold and separators grow as $O(|V|)$. In this case, we do not have a meaningful worst-case bound that can guarantee the efficiency of tensor network contraction. We hence resort to numerical experimentation.

# 3 Fast counting of solutions in #P-complete problems

## 3.1 Algorithms

### 3.1.1 Tensor network contraction algorithms

The algorithms used to prove the existence and the linear-time scaling for obtaining planar separators are not commonly implemented for graph partitioning in real-world problems. Furthermore, efficient partitioning of *arbitrary* graphs is crucial in a variety of fields, from network load balancing and power grid design to social and biological networks [44], and powerful methods have been developed for this purpose.

Here we employ the multilevel graph partitioning heuristic called METIS [45]. METIS performs one or more steps of coarse graining of the initial graph, in order to identify a favorable partition — that is, a partition that splits the graph into two induced subgraphs of roughly equal size by cutting as few edges as possible. METIS begins the partitioning procedure from a randomly chosen vertex. This random choice causes a small percentage of partition separators to be atypically long compared to the average for a particular graph class and size. We find that performing each partition twice and choosing the best out of 2 almost completely eliminates this shortcoming for the classes of graphs we have studied, with only minimal cost in computation time. We therefore always obtain 2 bipartitions of the same (sub)graph and choose the best one. For further details on the inner workings of METIS, we refer the reader to the relevant references.

The combined `tensor-METIS` algorithm works as follows. We begin by defining the tensor network corresponding to an instance of a #CSP problem, as detailed in Sec. 2.1. The first step of the algorithm is to construct a separator hierarchy of the network, like the one shown schematically in Fig. 2, by recursively performing METIS bipartition until each partition contains a single vertex (and hence a single tensor). We then use the separator hierarchy to determine the tensor contraction sequence, along the lines of the procedure described in the proofs of Corollaries 1 and 2. Carrying out all contractions yields the tensor trace, which is the count of satisfying assignments of the instance.

To provide another point of reference, we also devise a greedy contraction algorithm. At each step of this protocol, out of all connected pairs of tensors in the network, we choose the pair whose contraction yields the tensor with the smallest rank and contract it, and repeat this until the network is fully contracted. We call this heuristic `tensor-greedy`.

Finally, as an alternative to partitioning, one may consider the *community structure* of a network [46, 47] as a blueprint for efficient contraction sequences. Instead of separating a graph into a predetermined number of partitions, community detection algorithms automatically identify the number and membership of communities in a graph, such that vertices within a community are more highly connected than vertices across communities. Here we use the Girvan-Newman algorithm [48] as a heuristic to determine the tensor contraction sequence. The Girvan-Newman algorithm is based on *edge betweenness*, i.e., the number of shortest paths between pairs of vertices that contain an edge. The idea is that edges with high betweenness are more likely to connect vertices across communities, and thus recursively removing them eventually reveals the community structure in a graph. We use this community structure in the same way we use the separator hierarchies to contract arbitrary tensor networks. We designate this heuristic `tensor-GN`.

We have implemented the above tensor network contraction heuristics in Python and provide simple scripts demonstrating their usage on GitLab [49]. We use the python version of the igraph library to construct and manipulate graphs. Graph partitioning is implemented using the METIS library [45]. Finally, we use the igraph implementation of the Girvan-Newman algorithm in `tensor-GN`. All tensor contractions are performed using the library numpy. We

note that all the tensor contraction algorithms are exponential-space in the general case, and memory is the main performance bottleneck.

### 3.1.2 SAT counters used for benchmarking

It is unlikely that any #P-complete problem can be solved exactly in time that scales fundamentally better than exponentially [50]. Nevertheless, since solving #P-complete problems is often important for practical purposes, any benefit in performance can have far-reaching consequences. For this reason, copious effort has been invested in the last few decades in the development of highly optimized heuristics for the solution of #CSP problems [1], and in particular #SAT.

We compare our tensor network contraction algorithms against the fastest existing solvers for the solution of a class of #SAT problems, to be introduced below. Most solvers use the Davis–Putnam–Logemann–Loveland (DPLL) algorithm [51] to exhaustively search for all satisfying assignments of an instance, whereas others use ideas from knowledge compilation [52] to perform the counting. Specifically, we have tested the performance of `cachet` [53], `cnf2eadt` [54], `CNF2OBDD` [55], `d4` [56], `miniC2D` [57], `relsat` [58], and `sharpSAT` [59]. For knowledge compilers, we have used — whenever present — appropriate options to skip the compilation step and solely do counting. For more details on these algorithms, we refer to the respective references.

For the problems we study in the next Section, we find that `miniC2D` exhibits the best scaling of all the solvers we tested, followed by `d4`. For this reason, we will use `miniC2D` as a performance benchmark in Sec. 3.2. It is interesting to note that both `miniC2D` and `d4` perform a form of graph partitioning (more accurately, hypergraph partitioning) in an initial preprocessing step, which may play a role in their superior performance compared to other solvers in our experiments.

## 3.2 Numerical experiments

We have benchmarked our tensor network contraction algorithms on the #P-hard counting problems monotone #1-IN-3SAT and #CUBIC-VERTEX-COVER. Average-case hard instances of both these problems can be defined as random 3-regular (or cubic) graphs. This has the advantage that hard instances can be sampled almost uniformly, thus eliminating bias in their random selection. First we detail our results on monotone #1-IN-3SAT. Our results on #CUBIC-VERTEX-COVER are presented below.

Each monotone 1-IN-3SAT clause over three boolean variables returns TRUE if precisely *one* of the three variables is 1. The corresponding clause tensors have the form

$$T_{i_1 i_2 i_3}^{1\text{IN}3} = \begin{cases} 1, & \text{if } i_1 + i_2 + i_3 = 1 \\ 0, & \text{otherwise} \end{cases}. \tag{6}$$

Equivalently, a monotone 1-IN-3SAT clause over the variables $x_1, x_2, x_3$ can be written in conjunctive normal form as $(x_1 \lor x_2 \lor x_3) \land (x_1 \lor \neg x_2 \lor \neg x_3) \land (\neg x_1 \lor \neg x_2 \lor x_3) \land (\neg x_1 \lor x_2 \lor \neg x_3) \land \land (\neg x_1 \lor \neg x_2 \lor \neg x_3)$.

Monotone 1-IN-3SAT is NP-complete [60]. In fact, it represents the hardest type of satisfiability problems, as it features a "shattered" solution manifold close to the SAT-UNSAT threshold at clause-to-variable ratio $\alpha \simeq 2/3$. It is for this reason that this decision problem is used as a paradigmatic example to demonstrate the failure of all local search algorithms [61, 62]. Here we focus on precisely the regime where the decision problem becomes hard in the average case, but we instead tackle the counting counterpart, i.e., monotone #1-IN-3SAT, which is even harder than the decision problem. We generate random instances of monotone #1-IN-3SAT at $\alpha \simeq 2/3$, in which each variable appears in precisely two clauses. These instances can

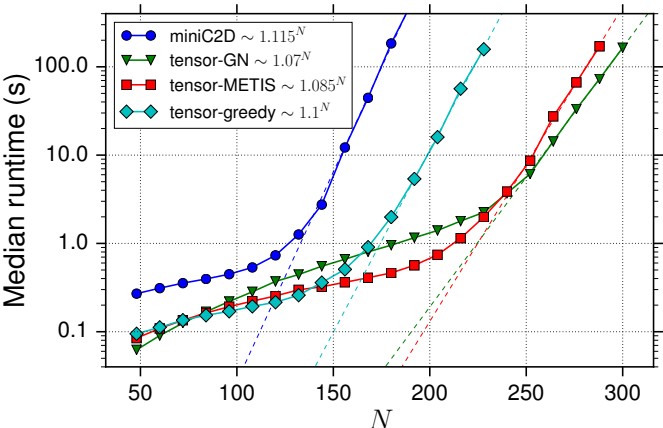

Figure 3: Median runtime scaling of tensor network contraction solvers and `miniC2D` for the solution of monotone #1-IN-3SAT on cubic graphs. All algorithms were run on the same random sample of 1000 instances of the problem. Dashed lines indicate the scaling extrapolated from the last 4 data points for each method. All calculations were performed on a single core of an AMD Opteron 6320 2800MHz processor with 64GB of RAM available.

be represented as 3-regular graphs, where each edge represents a variable and each vertex a clause. This allows us to efficiently sample them uniformly using established methods [63].

In Fig. 3 we compare the median runtime of `miniC2D` and our tensor methods as a function of the number of variables $N$ for 1000 random instances at each $N$. All our methods, including the naive greedy contraction algorithm, outperform `miniC2D`, achieving both better asymptotic scaling and absolute runtime. This exponential speedup allows us to reach instances of up to 300 variables with the graph partitioning and community structure heuristics, with the latter achieving a slightly better scaling than the former. This is well beyond what is achievable with all other known counting solvers. Our methods solve each individual instance in the sample faster than `miniC2D` in this regime. Note that, even though the median runtime is of the order of $10^2$ s for $N = 300$, exceptional instances are much harder, requiring $\sim 10^4$ s to solve.

Next, we apply our tensor network algorithms to the problem #CUBIC-VERTEX-COVER, i.e., counting the number of vertex covers of 3-regular graphs. A vertex cover is a subset of the vertices of the graph with the property that every edge of the graph has at least one of its endpoints in the subset. #CUBIC-VERTEX-COVER is #P-complete [64]. Equivalently, each instance of the problem can be defined as a #2SAT formula, in which a variable represents a vertex (so that $N = |V|$) and a two-variable OR clause represents an edge. We remark that, even though 2SAT is in P, even sampling satisfying 2SAT assignments uniformly or almost uniformly is hard unless P=RP [65], whereas the counting problem is #P-complete.

We can recast #CUBIC-VERTEX-COVER as a tensor network using Eqs. (3) and (4). Here we sample 1000 cubic graphs with 32-200 vertices randomly using a Monte Carlo procedure that guarantees asymptotic sample uniformity [63]. All the graphs we generate are guaranteed to be connected and simple, i.e., there are no multiple edges between vertices. Our results are summarized in Fig. 4. In Fig. 4(a) we present a comparison of median times to solution $\langle \tau \rangle$ between our tensor network contraction heuristics and the `miniC2D` algorithm. We observe that the exponential scaling of all the methods is revealed when instances grow beyond 100 vertices. We find that `tensor-METIS` outperforms `miniC2D` by more than an order of magnitude in runtime for the largest instances accessible. Even though the asymptotic scaling is

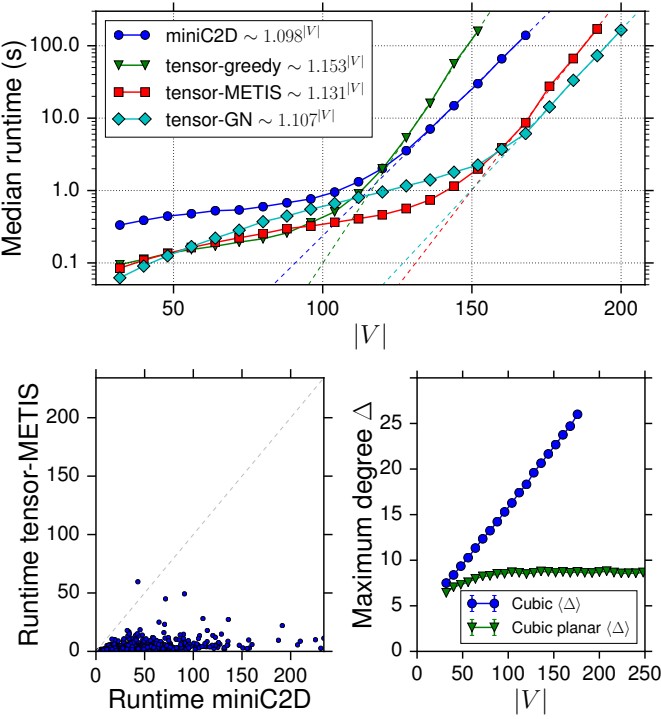

Figure 4: (a) Median runtime of tensor network contraction solvers and `miniC2D` for the solution of #CUBIC-VERTEX-COVER. All algorithms were run on the same random sample of 1000 instances of the problem. Dashed lines indicate the scaling extrapolated from the last 4 data points for each method. (b) Per instance runtime comparison between `tensor-METIS` and `miniC2D` for instances with 152 variables. Dashed gray line indicates equal runtime for the two algorithms. (c) Average maximum degree $\Delta$ encountered during the contraction sequence of `tensor-METIS` as a function of instance size for unrestricted (circles) vs planar (triangles) instances of #CUBIC-VERTEX-COVER. All calculations were performed on a single core of an AMD Opteron 6320 2800MHz processor with 64GB of RAM available.

slightly worse than that of `miniC2D`, an extrapolation of the exponential scaling shows that `tensor-METIS` is faster for all practically accessible instances of this problem: the curves for `tensor-METIS` and `miniC2D` meet for $|V| > 500$ and $\tau > 10^{25}$s, i.e., runtimes several orders of magnitude larger than the current estimate for the age of the universe. `tensor-GN` performs even better. Finally, Fig. 4(a) indicates that it is indeed the graph partitioning and community structure detection that are responsible for the good performance of the tensor algorithms, as it shows that `tensor-greedy` fails to achieve similar results.

Fig. 4(b) shows a more detailed comparison between runtimes of `tensor-METIS` and `miniC2D` for instances with 152 variables. Of the 1000 random instances of #CUBIC-VERTEX-COVER, only one is solved faster by `miniC2D`. Moreover, Fig. 4(b) shows that the vast majority of instances of this size are solved within 10 seconds by `tensor-METIS`, whereas the distribution for `miniC2D` is much broader. Finally, in Fig. 4(c) we show the growth of the average of the maximum degree encountered during the contraction sequence versus the number of variables, which reflects how the length of the separators constructed by METIS grows with $|V|$. The slope is clearly linear for general instances, indicating an exponential-time algorithm. As a comparison, we also show the maximum degree growth in solving planar instances of the same problem with `tensor-METIS`. This indicates that `tensor-METIS` is very effective in solving problems defined on planar graphs with degree constraints. We have observed the

same behavior in general random planar graphs [66].

We remark that in our numerical experiments, both tensor algorithms and `miniC2D` perform counting using fixed-precision floating-point arithmetic. For #CUBIC-VERTEX-COVER, rounding errors start occurring for $|V| \sim 100$ vertices, due to the fact that, for this problem, the number of solutions grows exponentially with $|V|$. This means that the counts we obtain from both methods are approximate. An exact count can be obtained by gradually incrementing the numerical precision upon increasing $|V|$. Increasing the representation size of numbers by a factor of $p$ means that the cost of a single multiplication of two numbers increases as $p^2$, and hence contraction of two tensors of total dimension $L$ now takes $O(p^2 L^2)$ multiplications. This means that extending the number representation can at most change the prefactor of the exponential scaling and not the exponent.

## 4 Summary and outlook

In this work, we have reformulated tensor network contraction as an algorithmic graph theory problem. In this language, it becomes apparent that there is a complexity dichotomy for tensor network contraction depending on whether the underlying graph possesses sublinear separators. This renders graph partitioning a potentially useful tool for finding favorable contraction sequences of tensor networks. We have verified that this is indeed the case by implementing tensor network contraction algorithms that demonstrably outperform established #SAT counters for instances of two #P-complete problems.

The techniques we develop have a number of limitations, all of which can be at least partly addressed. Since all of the argumentation depends only on graph properties, it is agnostic as to the computational problem embedded into the tensors. We have already demonstrated this fact by studying two different classes of problems on cubic graphs, of which the first (monotone #1-IN-3SAT) is considered harder to solve in practice. For example, the algorithms we develop here would have identical performance on #SAT and #XORSAT defined on the same graphs, even though the latter is a #-tractable problem. Another question pertains to graphs without upper bounded maximum degree. Vertex degrees correspond to tensor ranks, so if vertex degrees are allowed to grow linearly, then the size of the corresponding tensors grows exponentially. Nevertheless, it may be possible to relax the maximum degree condition in some cases. For SAT problems, one can break long clauses into two or more shorter clauses, incurring a polynomial overhead in ancillary variables, while tensors representing variables participating in many clauses can be exactly decomposed into rings of rank-3 tensors [21]. On the other hand, an increase in the average maximum degree is compensated by a decrease in the average clause-to-variable ratio. We thus expect that our algorithms should scale well for hard instances of #1-IN-$k$SAT with $k > 3$ as well, as the SAT-UNSAT threshold decreases with increasing $k$ for these problems [67].

In the tensor network representation of CSP problems, each variable and each clause correspond to a graph vertex. As our tensor network contraction methods scale with the number of vertices, increasing the clause-to-variable ratio multiplies the number of vertices in the graph and hence worsens the scaling of our methods in comparison to exhaustive DPLL solvers. Our methods are hence most advantageous for problems that are the hardest when the resulting instance graphs are sparser. A counterexample is #3SAT, which becomes hard for clause-to-variable ratios that are higher than those of the problems studied here. However, it is important to note that a higher threshold value of the clause-to-variable ratio does *not* necessarily signify a more complex problem: close to its satisfiability threshold at $\alpha \simeq 2/3$, monotone 1-IN-3SAT is practically a much harder decision problem than 3SAT close to its threshold at $\alpha \simeq 4.27$ [62]. In conclusion, even though our methods are applicable to a wide variety of CSPs, their poten-

tial is greatest for hard problems that give rise to sparser graphs, such as the aforementioned #1-IN-$k$SAT problems with $k > 3$.

Even though we have focused on #SAT problems, tensor network techniques can be used to solve CSPs beyond boolean satisfiability, including weighted satisfiability and CSPs with variables defined over higher dimensional — and even mixed — domains. We therefore believe this work may have important repercussions in a broad range of settings where fast solution of instances of #P-hard #CSP problems is required. For example, the evaluation of knot invariants can be thought of as weighted planar #CSP problems [68], for which our method of tensor network contraction is subexponential and demonstrably fast [66]. Furthermore, our approach can be used to classically simulate measurements of the outputs of random quantum circuits. The graph $G$ is then the circuit grid, tensors represent quantum gates, and the values of the variables representing the initial and post-measurement states of the circuit qubits are fixed. In this setting, a separator-based algorithm is equivalent — in terms of the functional form of the runtime scaling — to the established simulation algorithms based on treewidth [69], but may be advantageous in practice.

Another direction we believe holds promise is the generalization of tensor network compression techniques, developed for the coarse-graining of tensor lattices, to arbitrary graphs [70]. This could lead to novel subexponential-time approximation algorithms for decision and counting problems of intermediate complexity [71, 72]. Of particular future interest in this context is also combinatorial optimization.

Both graph partitioning and community detection lend themselves to divide-and-conquer schemes. With an efficient distributed representation of tensor networks [73], one could plausibly harness the potential of distributed computation over multiple machines or a cloud to solve very large instances of hard problems. Decision problems may be particularly amenable to such a strategy, if linear algebra of boolean tensors is implemented in a scalable manner.

Hybrid techniques that incorporate elements of more than one of the algorithms we introduced here may still improve performance. Tensor network contraction methods may hold even greater promise as members of algorithm portfolios, to be used in algorithm selection for artificial reasoning [74, 75]. Another appealing feature of tensor network contraction algorithms based on graph properties is that they can be "lazy", i.e., one can decide whether to evaluate the tensor trace after first contracting the underlying graph with a given method and tracking vertex degrees to accurately predetermine the computational cost of the tensor contraction — see, e.g., Fig. 4(c).

# Acknowledgements

We are grateful to K. Meichanetzidis, J. Reyes, Z.-C. Yang, and L. Zhang for many fruitful discussions. Parts of the computational work reported on in this paper was performed on the Shared Computing Cluster, which is administered by Boston University's Research Computing Services.

**Funding information**   S. K. was partially supported through the Boston University Center for Non-Equilibrium Systems and Computation. C. C. was partially supported by DOE Grant No. DE-FG02-06ER46316. E. R. M. was partially supported by NSF Grant No. CCF-1525943.

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
