# Peer review of "Fast counting with tensor networks"

_SciPost Physics, doi:SciPost Phys. 7, 060 (2019)_

## Round 1 · Referee Report · Anonymous (Referee 1) · 2019-8-25

Report

The paper seems to be written well and discussions are convincing. I recommend the publication.

---

## Round 1 · Referee Report · Anonymous (Referee 2) · 2019-10-17

Strengths

1- The paper is well-organized and well-written. 2- The paper is perfectly understandable for the tensor-network expert who is not familiar with CSPs, and (I assume) the other way around. 3- The benchmarks or "numerical experiments" are sound and the comparison with other state-of-the-art methods is convincing.

Weaknesses

none

Report

It would be good to clearly mention somewhere to what extent the two test cases are favourable to the tensor-network algorithms as compared to other techniques. The speed-ups that the authors found for these two cases, are these expected to be generic for a broad class of problems?

Requested changes

none

---

## Round 3 · Author Response

Dear Editor,
We thank you for considering our manuscript. We also thank the referees for their feedback on our work.
The second referee asked us to comment on the reasons for the superiority of our tensor network methods in solving the constraint satisfaction problems (CSPs) studied in this work and whether this superiority is expected to extend to a broad class of problems. Our response is that, at least in part, our tensor network methods benefit from the sparsity (i.e., small number of edges per vertex) of the networks that represent the instances of a CSP, which corresponds to instances with a lower clause-to-variable ratio α. We hence expect our methods to be advantageous for problems that are the hardest when the corresponding networks are sparser. This situation is not uncommon. For example, it encompasses all monotone #1-in-kSAT problems for k>=3. Even the decision counterparts of these problems close to their satisfiability threshold (α~2/3 for monotone 1-in-3SAT) are in fact much harder to solve than 3SAT close to its much higher threshold (α~4.27), due to the phenomenon of shattering discussed in Ref. [62] of the manuscript. This illustrates that our methods are relevant to a broad class of hard problems and a regime that is difficult to access with any other techniques.
We have added a paragraph in the discussion section (3rd paragraph in Sec. 4 of the revised manuscript attached) to address these points and hope that with this our paper is now in publishable shape.
Sincerely,
The authors
We thank you for considering our manuscript. We also thank the referees for their feedback on our work.
The second referee asked us to comment on the reasons for the superiority of our tensor network methods in solving the constraint satisfaction problems (CSPs) studied in this work and whether this superiority is expected to extend to a broad class of problems. Our response is that, at least in part, our tensor network methods benefit from the sparsity (i.e., small number of edges per vertex) of the networks that represent the instances of a CSP, which corresponds to instances with a lower clause-to-variable ratio α. We hence expect our methods to be advantageous for problems that are the hardest when the corresponding networks are sparser. This situation is not uncommon. For example, it encompasses all monotone #1-in-kSAT problems for k>=3. Even the decision counterparts of these problems close to their satisfiability threshold (α~2/3 for monotone 1-in-3SAT) are in fact much harder to solve than 3SAT close to its much higher threshold (α~4.27), due to the phenomenon of shattering discussed in Ref. [62] of the manuscript. This illustrates that our methods are relevant to a broad class of hard problems and a regime that is difficult to access with any other techniques.
We have added a paragraph in the discussion section (3rd paragraph in Sec. 4 of the revised manuscript attached) to address these points and hope that with this our paper is now in publishable shape.
Sincerely,
The authors

---

## Round 3 · List of Changes

Added paragraph in the discussion section to (i) address how graph sparsity affects the tensor network techniques introduced, and (ii) comment on the broadness of the class of problems accessible to these techniques.

---

## Editorial Decision

published